Phytoplankton diversity and chemotaxonomy in contrasting North Pacific ecosystems

Matek Antonija 1
Bosak Sunčica 1
Šupraha Luka 2 3
Neeley Aimee 4 5
Višić Hrvoje 6
Cetinić Ivona 4 7
Ljubešić Zrinka zrinka.ljubesic@biol.pmf.hr 1
1 Department of Biology, Faculty of Science, University of Zagreb , Zagreb , Croatia
2 Department of Earth Sciences, Uppsala University , Uppsala , Sweden
3 Section for Aquatic Biology and Toxicology (AQUA), Department of Biosciences, University of Oslo , Oslo , Norway
4 Ocean Ecology Laboratory, NASA/Goddard Space Flight Center , Greenbelt , MD , United States of America
5 Science Systems and Applications, Inc. , Lanham, Maryland , United States of America
6 Department of Geosciences, Faculty of Science, Eberhard Karls University Tübingen , Tübingen , Germany
7 GESTAR II, Morgan State University , Baltimore , MD , United States of America
Agusti Susana
Electronic publication date: 2023 Jan 3
Publication date: 2023
Volume: 11
Electronic Location ID: e14501
Received 2022 Jun 4; Accepted 2022 Nov 10
Copyright: ©2023 Matek et al.
Copyright year: 2023
Copyright holder: Matek et al.
License: This is an open access article distributed under the terms of the Creative Commons Attribution License, which permits unrestricted use, distribution, reproduction and adaptation in any medium and for any purpose provided that it is properly attributed. For attribution, the original author(s), title, publication source (PeerJ) and either DOI or URL of the article must be cited.
License URL: https://creativecommons.org/licenses/by/4.0/

Keywords: Phytoplankton taxonomy, Pigments, Trophic state, Particle abundance

Funding: Croatian Science Foundation UIP-2013-11-6433 IP-2020-02-9524 Schmidt Ocean Institute NASA GSFC Ocean Ecology Lab NASA’s PACE mission Uppsala University, Swedish Research Council grant 2011-4866 This work was funded by the Croatian Science Foundation under projects BIOTA (UIP-2013-11-6433), ISLAND (IP-2020-02-9524), the Schmidt Ocean Institute, NASA GSFC Ocean Ecology Lab, and NASA’s PACE mission. Jorijntje Henderiks provided financial support for SEM analyses at Uppsala University (Swedish Research Council grant 2011-4866 and other funding). The funders had no role in study design, data collection and analysis, decision to publish, or preparation of the manuscript.

==============================
Background

Phytoplankton is the base of majority of ocean ecosystems. It is responsible for half of the global primary production, and different phytoplankton taxa have a unique role in global biogeochemical cycles. In addition, phytoplankton abundance and diversity are highly susceptible to climate induced changes, hence monitoring of phytoplankton and its diversity is important and necessary.

Methods

Water samples for phytoplankton and photosynthetic pigment analyses were collected in boreal winter 2017, along transect in the North Pacific Subtropical Gyre (NPSG) and the California Current System (CCS). Phytoplankton community was analyzed using light and scanning electron microscopy and photosynthetic pigments by high-performance liquid chromatography. To describe distinct ecosystems, monthly average satellite data of MODIS Aqua Sea Surface temperature and Chlorophyll a concentration, as well as Apparent Visible Wavelength were used.

Results

A total of 207 taxa have been determined, mostly comprised of coccolithophores (35.5%), diatoms (25.2%) and dinoflagellates (19.5%) while cryptophytes, phytoflagellates and silicoflagellates were included in the group “others” (19.8%). Phytoplankton spatial distribution was distinct, indicating variable planktonic dispersal rates and specific adaptation to ecosystems. Dinoflagellates, and nano-scale coccolithophores dominated NPSG, while micro-scale diatoms, and cryptophytes prevailed in CCS. A clear split between CCS and NPSG is evident in dendogram visualising LINKTREE constrained binary divisive clustering analysis done on phytoplankton counts and pigment concentrations. Of all pigments determined, alloxanthin, zeaxanthin, divinyl chlorophyll b and lutein have highest correlation to phytoplankton counts.

Conclusion

Combining chemotaxonomy and microscopy is an optimal method to determine phytoplankton diversity on a large-scale transect. Distinct communities between the two contrasting ecosystems of North Pacific reveal phytoplankton groups specific adaptations to trophic state, and support the hypothesis of shift from micro- to nano-scale taxa due to sea surface temperatures rising, favoring stratification and oligotrophic conditions.

Introduction

Phytoplankton have many important roles in the marine ecosystem: they are responsible for half of the global primary production (Chavez, Messié & Pennington, 2011), contribute to the biogeochemical cycles by being part of the biological pump through nutrient uptake and carbon sequestration (Volk & Hoffert, 1985; Michaels & Silver, 1988; Karl & Church, 2017), and they are at the base of the majority of ocean ecosystems (Pomeroy, 1974; Sherr & Sherr, 1991; Legendre & Le Fèvre, 1995). Therefore, any changes in phytoplankton diversity impact the oceanic carbon cycle, nutrient uptake, and zooplankton community structure, which has an indirect effect on the whole oceanic ecosystem (Ramond et al., 2021). Consequences of global warming such as temperature increase, change in ocean circulation and stratification, acidification, and deoxygenation have an impact on the phytoplankton community (Rost, Zondervan & Wolf-Gladrow, 2008). It is predicted that increases in ocean temperature and other climate induced changes will affect phytoplankton metabolic rates and growth, ultimately changing the ocean-wide phytoplankton diversity, and overall marine productivity (Moore et al., 2018; Cael, Dutkiewicz & Henson, 2021). Due to this expected change in phytoplankton community in the oceans of tomorrow, it is of extreme importance to understand the current oceanic phytoplankton diversity and how it is shaped by environmental factors.

The North Pacific ecosystem is influenced by the trade winds, the anticylonic North Pacific Subtropical Gyre (NPSG), and the cyclonic Subarctic Gyre that bifurcate into the California Current System (CCS) and Alaska Current. The CCS is a transitional ecosystem that is more eutrophic in comparison to NPSG because of the Columiba River’s contributon of terrigenous sediments and organic matter (Kammerer, 1987; Morgan, De Robertis & Zabel, 2005; Steele, Thorpe & Turekian, 2008; Kudela et al., 2010; Capone & Hutchins, 2013; Catlett et al., 2021; Closset et al., 2021; Abdala et al., 2022). Phytoplankton diversity of the North Pacific was well recorded in literature of 1970–1990s at one station, defined as CLIMAX area (28°N, 155°W) (Venrick, 1997). The community was dominated by diatoms, then equally comprised of dinoflagellates, and coccolithophorids, while cryptophytes, chryosphyceae, cyanobacteria and other groups contributed less (Venrick, 1971; Venrick, 1982; Hayward, Venrick, McGowan, 1983; Venrick et al., 1987; Venrick, 1990. Species number varied between 100 and 300, depending on number of samples and ability to identify nano-fraction (Venrick, 1982; Venrick et al., 1987; Venrick, 1990). Recent research shows oligotrophic areas of North Pacific are usually dominated by pico- and nanophytoplankton (Karl & Church, 2017; Kodama et al., 2021), while high community diversity, with presence of larger microphytoplankton (e.g., diatoms) is found in eutrophic regions of North Pacific (Du, Peterson & O’Higgins, 2015; Du & Peterson, 2018).

Large oceanic ecosystems, such as the North Pacific Ocean are showing response to changes in climate (Venrick et al., 1987; Bograd et al., 2019; Litzow et al., 2020). For instance, in autumn of 2013, a warm blob appeared in the Gulf of Alaska, and by December of 2015, it expanded toward the Bering Sea, the Transition Zone, and the California Current System (CCS) (Peterson, Bond & Robert, 2016). The blob-induced increase of the sea surface temperatures had an effect on ecosystem, especially phytoplankton community structure across the whole North-East Pacific. Phytoplankton community of oligotrophic NPSG shifted from nanophytoplankton to picophytoplankton during warm phases of climate oscillations when stratification is strong, and particle export is low (Yoon & Kim, 2020). Moreover, during the warm blob anomaly in the eutrophic and diatom-dominated CCS, nutrient supply decreased for 50% and the phytoplankton community shifted to nonsiliceous phytoplankton and/or lightly silicified diatoms (Closset et al., 2021). Long data records for North Pacific are collected at station ALOHA (22.75°N, 158°W: A Long-term Oligotrophic Habitat Assessment) in NPSG (Karl & Church, 2017) and Station M (34°50′N, 123°00′W; 4,000 m depth) in CCS (Monterey Bay Aquarium Research Institute, 2022). Three decades of data from ALOHA combined with improved satellite algorithms are showing different trends of phytoplankton biomass, and net primary production growth in response to positive phases of the North Pacific Gyre Oscillation, Pacific Decadal Oscillation and El Niño Southern Oscillation (Kavanaugh et al., 2018). Furthermore, two-decade record on abyssal ecosystem at station M show strong benthic-pelagic cupling, and significant response of the benthic communities to the climate induced changes in the ocean surface (Monterey Bay Aquarium Research Institute, 2022). All these studies demonstrate the importance of time-series studies to record and predict future changes in ecosystems.

Recent advances in molecular and imaging technologies offer an unprecedented view of the oceanic diversty (Gorocs et al., 2018; Shin et al., 2018; Fender et al., 2019; Hoving et al., 2019; Fischer et al., 2020; Mirasbekov et al., 2021; Clayton et al., 2022). In a similar way, chemotaxonomy offers the additional insight into the phytoplankton. However, our vision of the phytoplankton diversity still relies on the morphological community structure and direct connection with remote sensing (Kramer et al., 2022) characterisation, usually done by imaging. Image based taxonomy, although time consuming is by far the most wide-spread method in determining phytoplankton community structure, in addition thanks to the new automated instruments and technologies (Olson & Sosik, 2007; Picheral et al., 2010; Clayton et al., 2022). Advances in imaging technology are contrasted by the decline in numbers of highly trained taxonomic analysts as well as the new trainees entering the pipeline (Drew, 2011; Pearson, Hamilton & Erwin, 2011; McQuatters-Gollop et al., 2017; Orr et al., 2021; Clayton et al., 2022).

To fully understand the Pacific ecosystem, it is necessary to develop knowledge of the phytoplankton diversity that relates to different ecosystems, changes in environment, and that can be used for future predictions of global warming’s impact on marine ecosystems. Therefore, the aim of this research was to represent a phytoplankton diversity of distinct trophic systems in North Pacific using microscopy counts as main method, in combination with chemotaxonomy, on a large spatial scale in a short period of time.

Materials & Methods

Expedition- location and time

A Sea to Space Particle Investigation cruise aboard the Schmidt Ocean Institute R/V Falkor was conducted from January 24 to February 20, 2017, in North Pacific (Fig. 1). The aim of the cruise was to connect the radiometric properties (ocean colour) as well as ecological mechanisms of carbon export (Durkin et al., 2022) with the trophic state of the ocean, and use those data to develop algorithms and phytoplankton proxies for the NASA’s PACE mission (pace.oceansciences.org).

Figure 1 Investigation area superimposed on satellite data.

Cruise track of the Sea to Space cruise (black line), showing approximate position of Station 1, Station 2, and Station 3, superimposed onto: (A) MODIS Aqua Sea Surface temperature, (B) MODIS Aqua Chlorophyll a concentration, and (C) Apparent Visible Wavelength. All satellite data is monthly average for February 2017.

Sampling

Sampling was done along the investigated transect at Station 1 (ST1) and Station 2 (ST2) in NPSG, and Station 3 (ST3) in CCS (Fig. 1). Each station represents a group of sampling sites (Table 1) where CTD (conductivity, temperature, depth) casts were deployed at three depths: the surface layer (S), deep chlorophyll maximum (DCM), and mixed layer depth (MLD), with exception at CTD 14 where additional sample was taken below mixed layer depth (BMLD) (Table 1). Due to the strong physical forcing, water column was well mixed as shown in CTD profiles of all three stations in Durkin et al. (2022) (Fig. S1). Samples for phytoplankton (n = 38) and pigment analyses (n = 38) were taken by 10 L Niskin rosette sampler equipped with CTD and other sensors. For more detailed taxonomic analyses, additional samples (n = 38) were taken from the same Niskin bottles, and volume of 400 mL seawater was filtered using weak vacuum onto 0.8 µm polycarbonate filters analyzed on SEM as described in (Šupraha, Ljubešić & Henderiks, 2018).

Table 1 Sampling sites within each station: Station 1 (ST1), Station 2 (ST2), and Station 3 (ST3).

Total number of samples (N)=103, of which N = 25 at ST1, N = 22 at ST2, and N = 56 at ST3. CTD casts, corresponding depths and water column layers are shown for each site, as well as which sample type is taken (+). Abbreviations: PHYTO (samples (N = 38) taken for light microscopy and pigment analyses); SEM (samples ( N = 38) taken for scanning electron microscopy); NET (samples ( N = 27) taken with phytoplankton net with 20 µm mash size).

Station	Sampling site (latitude; longitude)	CTD Cast	Depth	Water column layer	PHYTO	SEM	NET	
Station 1	22°14.6892; −151°52.2906	CTD4	0	S	+	+		
CTD4	115	DCM	+	+		
CTD4	130	MLD	+	+		
22°16.5251; −151°44.8940	CTD8	0	S	+	+		
CTD8	115	DCM	+	+		
CTD8	125	MLD	+	+	+	
22°16.5251; −151°44.8940	CTD14	0	S	+	+	+	
CTD14	88	DCM	+	+	+	
CTD14	128	MLD	+	+	+	
CTD14	180	BMLD	+	+	+	
Station 2	27°42.5971; −139°29.9381	CTD18	0	S	+	+	+	
CTD18	98	DCM	+	+	+	
CTD18	128	MLD	+	+	+	
27°39.6715; −139°33.0614	CTD19	130	MLD		+		
CTD19	composite				+	
27°42.0327; −139°41.7295	CTD21	0	S	+	+		
27°44.7694; −139°40.2311	CTD22	0	S	+	+	+	
CTD22	95	DCM	+	+	+	
CTD22	120	MLD	+	+	+	
Station 3	34°34.1060; −123°30.6151	CTD29	0	S	+	+		
CTD29	31	DCM	+	+		
CTD29	42	MLD	+	+		
CTD29	composite				+	
	CTD33	0	S	+	+		
34°31.5869; −123°33.9840	CTD33	27	DCM	+	+		
	CTD33	30	MLD	+	+		
	CTD33	composite				+	
	CTD37	0	S	+	+		
34°18.2352; −123°32.4584	CTD37	2	DCM	+	+		
	CTD37	38	MLD	+	+		
	CTD37	composite				+	
	34°30.011; −123°11.1985	CTD38	0	S	+	+	+	
	34°54.3259; −122°41.4444	CTD39	0	S	+	+	+	
	35°40.1678; −121°55.7237	CTD40	0	S	+	+	+	
	35°58.2849; −122°13.5212	CTD41	0	S	+	+	+	
		CTD41	14	DCM	+			
		CTD41	27	MLD	+	+		
		CTD41	composite				+	
	41°28.4439; −126°18.8841	CTD42	0	S	+	+	+	
	41°30.6406; −125°20.7072	CTD43	0	S	+	+	+	
	CTD43	80	DCM	+	+	+	
	CTD43	90	MLD	+	+	+	
	CTD43	composite				+	
	41°32.8395; −124°24.2721	CTD44	0	S	+	+	+	
	41°32.8395; −124°24.2722	CTD46	0	S	+	+	+	

For qualitative plankton analysis, another set of samples (n = 27) was taken from the Niskin bottles and filtered through 20 µm mesh. Discrete phytoplankton and net phytoplankton samples were fixed with 2% neutralized formaldehyde and stored in 250 mL bottles until analyses in the Laboratory of Biological Oceanography, Department of Biology, University of Zagreb. Triplicate 4 L seawater samples were filtered on GF/F filters for phytoplankton pigment analysis and stored in liquid nitrogen until the high-performance liquid chromatography (HPLC) analysis in the NASA’s Goddard Space Flight Center, following methods described in Hooker et al. (2012).

Phytoplankton community analysis

Light microscopy (LM) was used to determine phytoplankton composition and abundance. Subsamples of 50 or 100 mL, depending on cell density, were settled for 24 h and 48 h respectively and analyzed under a Zeiss Axiovert 200 inverted microscope using the Utermöhl method (Utermöhl, 1958) as described in (Šupraha et al., 2016). For additional taxonomic analyses, net samples were analyzed with the Zeiss Axiovert 200 inverted microscope and images of all species were taken and analyzed with Zeiss AxioVision SE64 (version 4.9.1). Micrograph plate of dominant taxa was made and edited using Adobe’s Photoshop CC 2015 and Illustrator CC 2017.

Phytoplankton are comprised of a phylogenetically diverse group of both prokaryotic and eukaryotic organisms. Because of that, classification is much debated with different systematic grouping (Bray & Curtis, 2006; Roy et al., 2011; Thomas et al., 2012; Pal & Choudhury, 2014). Therefore, a simpler approach for classification will be presented in this paper with focus on morphological characteristics of most abundant forms only: cyanobacteria, diatoms, dinoflagellates, coccolithophores, cryptophytes and “others”–including phytoflagellates, silicoflagellates, ciliates and other genera. Phytoplankton were classified on size variation using the equivalent spherical diameter (ESD) of cells as nanophytoplankton (ESD 2–20 µm) and microphytoplankton (ESD 20–200 µm).

Trophic indices and spatial distribution

Pigment average concentrations were calculated in order to get Fp index using formula by Claustre (1994): Fp = (sum of average concentrations of fucoxanthin and peridinin)/(sum of average concentrations of all primary pigments). Spatial distribution across investigated transects was visualized by creating one chart showing abundances of phytoplankton groups, and another one with distribution of the subset of pigments that best correlate to phytoplankton community (the correlation test explained later in Statistical analysis section). Chart plotting and calculations were made using the software Grapher 12 (GoldenSoftware) and Microsoft Office 365 ProPlus (Microsoft Corporation, version 1705), respectively.

Satellite data

To illustrate distinct trophical ecosystem-specific properties of investigated area, monthly average satellite data (February 2017) of MODIS Aqua Sea Surface temperature and Chlorophyll a concentration, as well as Apparent Visible Wavelength (AVW) was used. AVW is an optical water type classification that allows for a single, highly sensitive metric to combine the information about the spectral shape of the ocean colour (Vandermeulen et al., 2020), where spectral shift in AVW indicates changes in oceanic components contribution to the ocean color. In open ocean, ocean colour is driven by the phytoplankton community and associated detrital component, and in coastal ocean dissolved organic matter and sediment can contribute as well. As such, it is a great geophysical tool to evaluate spatial and temporal changes across oceanic ecosystems.

Statistical analysis

Several statistical analyses were done using Primer 7.0. (2021; PRIMER-E, Ltd., Auckland, New Zealand) to test similarities between ST1, ST2 and ST3, and correlation between phytoplankton counts and pigment data in order to gain better understanding of community diversity.

Analysis of similarity

Bray–Curtis (BC) rank similarity matrix was calculated using log(x+1) transformed data (Bray & Curtis, 2006) of phytoplankton counts. To test significance of similarity between ST1, ST2, and ST3, we run pairwise analysis of similarity (ANSOIM R statistic) on BC rank similarity matrix. Test takes averages of ranks within matrix and calculates their differences within each group in the cluster (Clarke et al., 2014). Furthermore, similarity percentages analyses (SIMPER) (Clarke, 1993) were used to observe the percentage contribution of each taxon to the average dissimilarity between samples of different groups (ST1, ST2, and ST3).

Correlation tests

In addition, another BC rank similarity matrix was calculated on log(x+1) transformed data of pigment concentrations at ST1, ST2, and ST3. We run RELATE analysis, BEST global test, and LINKTREE analyses using both BC matrices in order to test if there is a significant correlation between pigment concentrations and phytoplankton counts data.

RELATE statistic with Spearman correlation method shows how well two similarity matrices relate to each other by calculating correlation factor (Clarke et al., 2014). The analysis was done on BC rank similarity matrix of pigments concentrations and BC matrix of phytoplankton counts. In case RELATE analysis indicate a high correlation factor, BEST global test is run to find the subset data of one BC matrix (in our case pigment concentrations) that explains the structure of data in another BC matrix (in our case phytoplankton counts) (Clarke et al., 2014). In that way we aim to calculate which set of pigments have the highest correlation with the phytoplankton community structure.

In order to visualize the correlation between resulted pigment set and phytoplankton counts, and test its significance, LINKTREE constrained binary divisive clustering analysis and similarity profile test (SIMPROF) were run, respectively (Clarke et al., 2014). LINKTREE produces a dendrogram that shows clustering of ST1, ST2, and ST3 based on phytoplankton counts, and at the same time explains the cluster structure by showing which pigment concentration thresholds cause the main splits.

Results

North Pacific ecosystem and water column hydrography

Satellite data monthly averaged for February 2017 confirm that ST1, ST2 (NPSG), and ST3 (CCS) are located in distinct ecosystems (Fig. 1). NPSG has higher sea surface temperature compared to CCS (MODIS Aqua Sea Surface temperature data, Fig. 1A), where Chl a concentration is much higher (MODIS Aqua Chlorophyll a concentration data, Fig. 1B). Furthermore, MODIS Aqua Apparent Visible Wavelength data (Fig. 1C) indicates differences between two sampled environments based on the spectral shape of the color of the ocean: eutrophic CCS (higher AVW) and oligotrophic NPSG (lower AVW).

The deep chlorophyll maximum layer (DCM) was defined as highest fluorescence signal encountered during station profiles. For profiles collected at ST1 and ST2, it is set at ∼130 m, while it was found at much shallower depths at coastal ST3 (∼30 m). As expected, mixed-layer depth (MLD, calculated as the depth at which density differed from the mean density in the top 10 m by <0.05 kg m−3), was sitting in proximity of the DCM, at ∼130 m for ST1 and ST2, and at ∼90 m depth at ST3.

Phytoplankton diversity of North Pacific Ocean

The encountered phytoplankton community was mostly comprised of coccolithophores (35.5%), diatoms (25.2%) and dinoflagellates (19.5%) while cryptophytes, phytoflagellates and silicoflagellates, etc. were included in group “other” that makes 19.8% of phytoplankton counts. A total of 207 taxa have been determined from both Niskin and net samples of which: 106 diatoms, 48 coccolithophores, 41 dinoflagellates, seven other autotrophs, four heterotrophs, and one cyanobacterium. Cryptophytes were observed but were not identified to the genus level (Table S1). Of the 207 taxa, more than a half (113) taxa are found only in net samples: 42 diatoms, 40 coccolithophores, 27 dinoflagellates and four other heterotrophs.

Spatial distribution of phytoplankton groups using microscopy and pigments

Microscopy counts resulted in abundances of phytoplankton groups that indicate lower biomass of micro- and nanophytoplankton at NPSG oligotrophic ecosystem (ST1 and ST2) in comparison to eutrophic CCS (ST3). Moreover, results elucidate variable spatial distribution of microphytoplankton, while spatial distribution of nanophytoplankton is even (Figs. 2A and 2B). Diatoms of micro-fraction increased for an order of magnitude with the transition to ST3, while distribution of dinoflagellates, coccolitophores, and other phytoplankton groups of microphytoplankton stay constant across the investigated transect (Fig. 2A). Nano fraction of diatoms, dinoflagellates, and coccolitophores had even distribution across stations, while “other” cells (e.g., cryptophytes) exhibited similar behaviour to micro-scale diatoms, increasing their abundances at ST3 (Fig. 2B).

Figure 2 Spatial distribution of phytoplankton along the sampling transect in North Pacific.

(A) microphytoplankton fraction; (B) nanophytoplankton fraction. Stations (Station 1, Station 2, and Station 3) with sampling sites as CTD casts and corresponding depth (the surface layer (S), deep chlorophyll maximum (DCM), and mixed layer depth (MLD)) are shown on x-axis. Abundances (cellsL−1) of diatoms, dinoflagellates, coccolithophores, and others are shown on y-axis.

Average pigment concentrations encountered on transect (Table S4) show Fp index that is higher at ST3 (0,087), and lower at ST1 (0,018) and ST2 (0,021). Alloxanthin, zeaxanthin, divinyl chlorophyll b (DVChl b), and lutein are the pigment set with the highest correlation to phytoplankton counts, as identified by the BEST global test that resulted in Spearman correlation coefficient (Rho = 0.532) with p <0.1% significance level (Table S3). Spatial distribution of these four pigments and divinyl chlorophyll a (DVChl a) across stations elucidates two clearly distinct environments (Fig. 3A). ST1 and ST2 exhibited higher concentrations of DVChl a, and zeaxanthin, the biomarkers for Prochlorococcus and Synechococcus (respectively), implying the cyanobacteria domination in this region. Entering ST3, concentrations of previous pigments fall substantially, while concentrations of cryptophytic biomarker alloxanthin rise. Moreover, we observed higher concentrations of 19′-hexanoyloxyfucoxanthin (19HF) and fucoxanthin (Table S4), biomarkers for coccolithophores and mostly diatoms, respectively. Biomarkers peridinin and prasinoaxanthin also dominated at ST3, representing high abundances of dinoflagellates and prasinophytes, respectively (Table S4). Furthermore, there is a strong increase of total Chl a concentration at ST3, when compared to oligotrophic ST1 and ST2 (Fig. 3B).

Figure 3 Spatial distribution of pigments along the sampling transect in North Pacific.

(A) Pigments that correlated the most with the phytoplankton abundances: alloxanthine, zeaxanthin, divinyl chlorophyll a (DVChl a), divinyl chlorophyll b (DVCHl b), and lutein. (B) Total chlorophyll a (Chl a). Stations (Station 1, Station 2, and Station 3) with sampling sites as CTD casts and corresponding depths (the surface layer (S), deep chlorophyll maximum (DCM), and mixed layer depth (MLD)) are shown on x-axis. Concentrations of pigments (µgL −1) are shown on y-axis (log-scale at (b)).

Similarity between stations and dominant taxa

Pairwise test of ANOSIM analysis displayed significant differences in phytoplankton community abundance and composition between ST1 and ST3, and ST2 and ST3 with R-value being 0.579 and 0.612, respectively (Table S2). Taxa diversity and abundances were largest at eutrophic ST3, while oligotrophic ST1 and ST2 exhibited similar community structure.

Nano-scale dinoflagellates and coccolithophores contributed the most to dissimilarity of both ST1 and ST2 according to SIMPER analysis results (Table 2). Next were phytoflagellates with high contribution to the ST1 dissimilarity, while at ST2 that was Rhizosolenia hebetata f. semispina (Table 2). Nano-scale coccolitophorids (ESC 5–10 µm), cryptophytes and Pseudo-nitzschia pseudodelicatissima contributed the most to dissimilarity of ST3 (Table 2).

Table 2 Similarities percentage (SIMPER) analysis for each taxon/group by stations ST1, ST2, and ST3.

Analyses was done on samples for light microscopy (N = 38) and net phytoplankton samples (N = 27), of which N = 15 at ST1, N = 14 at ST2, and N = 36 at ST3. Blank cells are values that could not be determined because there were less than 40 cells in 1L. Taxa with similarity contribution <2 have been excluded from this table. Abbreviations: average contribution/standard deviation (δ/ σ), species contribution (Σδ%).

Taxon/Group	Station 1 (δ/σ, Σδ%)	Station 2 (δ/σ, Σδ%)	Station 3 (δ/σ, Σδ%)	
Undetermined dinoflagellates (10–20 µm)	5.44, 13.84	8.06, 12.37	1.64, 7.41	
Undetermined coccolitophorids (<5 µm)	1.79, 12.09	6.72, 14.02	1.67, 7.81	
Cryptophyceae	0.91, 5.75	1.02, 5.60	4.97, 9.02	
Gyrodinium spp.	0.65, 2.50	0.70, 2.05	1.26, 4.02	
Nitzschia bicapitata	0.52, 2.50	0.68, 2.45	0.65, 2.10	
Chaetoceros perpusillus	0.61, 2.39	1.03, 3.22		
Undetermined dinoflagellates (5–10 µm)	0.91, 6.13	0.72, 4.30		
Leptocylindrus mediterraneus	0.66, 2.02	0.72, 2.05		
Undetermined coccolitophorids (5–10 µm)	8.47, 15.61		6.40, 10.06	
Nitzschia longissimi	0.67, 2.33		2.17, 5.41	
Phytoflagellates	1.24, 7.61		1.06, 4.63	
Undetermined pennate diatoms (<20 µm)	0.53, 2.36		0.75, 3.16	
Nitzschia sp.	0.67, 2.06			
Michelsarsia adriatica	0.63, 2.37			
Gyrodinium spp. (<20 µm)	0.53, 2.54			
Gymnodinium spp.	0.52, 2.67			
Undetermined coccolithophorids (10–20 µm)		7.43, 12.20		
Discosphaera tubifera		1.64, 6.67		
Calciosolenia murrayi		1.55, 5.87		
Calciosolenia brasiliensis		0.72, 2.99		
Rhizosolenia hebetata f. semispina		7.34, 14.73	1.60, 3.52	
Pseudo-nitzschia pseudodelicatissima			6.14, 7.62	
Chaetoceros convolutes			1.61, 4.93	
Rhizosolenia cleveii			1.03, 2.68	
Lennoxia faveolata			0.88, 3.84	
Thalassiosira (<20 µm)			0.88, 3.83	
Chaetoceros contortus			0.74, 2.21	

Dominant taxa by stations were defined as species and groups with abundance >104 cells L−1, and the frequency of occurrence in samples >50% (Table 3), and some of them are shown on micrographs (Fig. 4). Dominant taxa present along the whole transect, but reaching highest abundances at ST3 were cryptophytes, Gyrodinium spp, Nitzschia bicapitata, nano-dinoflagellates and nano-coccolithophores (ESC <10 µm) (Table 3). On the other hand, some dominant taxa were present at only one station. Species specific to ST1 were nano-scale Gyrodinium sp. (ESC <20 µm), Gymnodinium spp., Michaelsarsia adriaticus, N. braarudii, and Nitzschia sp. Specific taxa at ST2 were Calciosolenia brasiliensis, Nitzschia sp., Ophiaster sp., and nano-coccolithophores (ESC 10–20 µm). The highest number of specific species was found at ST3, and most of them were diatoms: Chaetoceros contortus, Ch. convolutus, Ch. debilis, Lennoxia faveolata, N. sicula, Proboscia alata, Pseudo-nitzschia pseudodelicatissima, R. hebetata f. semispina, R. cleveii, Thalassionema nitzschioides, and nano-scale Thalassiosira sp. (ESC <20 µm). Other two specific taxa for ST3 were Micromonas sp. and Oxytoxum cf. variabile (ESC <20 µm) (Table 3). Highest abundance of diatoms at ST3 was recorded thanks to the high quantities of Pseudo-nitzschia pseudodelicatissima.

Table 3 Maximum abundances (cells L−1), and frequencies (%) for dominant species (where dominance is defined as frequency of appearance in samples >50%) at Station 1 (ST1), Station 2 (ST2) and Station 3 (ST3).

Total number of samples (N)=103, of which N = 25 at ST1, N = 22 at ST2, and N = 56 at ST3. Blank cells are values that could not be determined because there were less than 40 cells in 1 L.

Dominant Taxa/Group	Max (ST1)	Fr (ST1)	Max (ST2)	Fr (ST2)	Max (ST3)	Fr (ST3)	
Chaetoceros contortus					2660	63	
Chaetoceros convolutus					5320	88	
Chaetoceros debilis					2660	50	
Chaetoceros perpusillus	380	60	380	75			
Lennoxia faveolata					14200	69	
Leptocylindrus mediterraneus	190	60	380	63			
Nitzschia bicapitata	710	50	1420	63	4260	56	
Nitzschia braarudii	190	50					
Nitzschia longissima	285	60			3800	94	
Nitzschia sicula					760	50	
Nitzschia sp.	570	60					
Nitzschia sp. 1			285	50			
Proboscia alata					380	50	
Pseudo-nitzschia pseudodelicatissima					22420	100	
Rhizosolenia hebetata f. semispina					1900	88	
Rhizosolenia cleveii					1140	75	
Thalassionema nitzschioides					1900	50	
Thalassiosira sp. (<20 µm)					8520	69	
Unknown diatoms (<20 µm)	1420	50			10650	63	
Gymnodinium spp.	380	50					
Gyrodinium spp.	710	60	190	63	1140	81	
Gyrodinium spp. (<20 µm)	3550	50					
Oxytoxum cf. variabile (<20 µm)					2130	50	
N.D. dinoflagellates (5–10 µm)	1420	70	2130	63	19880	50	
N.D. dinoflagellates (10–20 µm)	2840	100	4615	100	19880	88	
Calciosolenia brasiliensis			380	63			
Calciosolenia murrayi	570	50	760	88			
Discosphaera tubifera	570	50	760	88			
Michaelsarsia adriaticus	190	60					
Ophiaster sp.			950	50			
N.D. coccolithophorids (<5µm)	3550	90	7810	100	24140	88	
N.D. coccolithophorids (5–10 µm)	4615	100	8520	100	29820	100	
N.D. coccolithophorids (10–20 µm)			3195	100			
Cryptophyceae	1065	70	1065	75	32660	100	
Micromonas sp.					2840	50	
Phytoflagellates	1065	80			8520	75	

Figure 4 Micrographs of dominant species at Station 1 (ST1), Station 2 (ST2) and Station 3 (ST3).

From top left to bottom right: (A) Chaetoceros convolutus (ST3), (B) Rhizosolenia clevei with Richelia intracelularis (arrow, ST3), (C) Nitzschia longissima (ST1), (D) Thalassiosira sp. (ST3), (E) Ophiaster sp. (ST2), (F) Cryptophyta (ST3), (G) Phytoflagellates (ST1), (H) Chaetoceros debilis (ST3), (I) Thalassionema nitzschioides (ST3), (J) Michaelsarsia adriaticus (ST1), (K) Nitzschia bicapitata (ST3), (L) Discosphaera tubifera (ST2).

Correlation between pigments and phytoplankton counts

HPLC based pigment concentrations closely followed the significant across-transect trends observed in phytoplankton abundances, as demonstrated by the RELATE test (Fig. S1B). Alloxanthin, zeaxanthin, DVChl b, and lutein contributed the most to similarities in trends, as shown by the BEST global test (Table S3). A clear split between coastal, eutrophic ST3 and oligotrophic ST1 and ST2 is visible in dendogram visualising LINKTREE constrained binary divisive clustering analysis done on phytoplankton counts and pigment concentrations (Fig. 5). This primary split (Node A>B, K), that can be explained by the specific threshold of alloxanthin (−0.0089 µgL−1 for ST1 and ST2, 0,001 µgL−1 for ST3) is highly significant (SIMPROF test, Fig. S1A).

Figure 5 LINKTREE binary divisive clustering analysis of the phytoplankton community at 37 sites.

Each split is constrained by a threshold of one of four best correlated pigments: alloxanthin (Allo), zeaxanthin (Zea), divinyl Chl b (DVChl b), and lutein (Lut). The first in-equality indicates sites to the left side of the split, the second sites to the right. The primary split is marked with A. Clusters marked with red dotted line are not significant by SIMPROF test. Split results: A- >B,K Allo < − 8, 89E +03 (>0,001); B-> C Lut < −8,89E +03 (>0,001) or Zea <0,065 (>0,111); K- >L,N Zea <0,007 (>0,012); N-> O Lut <0,007 (>0,01).

Further splits in the dendogram are driven by secondary pigments and demonstrate finer differences within the ecosystem types, on oligotrophic side lutein or zeaxantin (Node B), and on eutrophic side by zeaxantin (Node K>L, N) and further down (Node N) by the lutein. Note that only some of the splits in this dendogram are significant (black lines on the Fig. 5) according to SIMPROF test (Fig. S1A).

Discussion

Horizontal distribution of phytoplankton

Planktonic dispersal rate varies across marine planktonic taxa, while negative relationship between dispersal scale and body size causes less abundant and larger-fraction plankton (in near-surface, epipelagic waters) to have shorter dispersal scales and larger spatial species-turnover rates than the more abundant, smaller-fraction plankton (Villarino et al., 2018). The larger phytoplankton will be more similar at geographically proximate locations, and dissimilar between distant locations while it would allow smaller, more abundant phytoplankton (body size <2 mm) to travel greater distances (Finlay, 2002; Martiny et al., 2006; Villarino et al., 2018). This explains even spatial distribution of all nanophytoplankton fractions between stations, while microphytoplankton fractions, especially diatoms, are most abundant at ST3 (Figs. 3A and 3B). Additionally, we observed the highest number of specific diatom species at ST3 (Table 2).

Phytoplankton community structure

Phytoplankton community of North Pacific was comprised of 207 taxa, of which most abundant were coccolithophores (35.5%), then diatoms (25.2%), dinoflagellates (19.5%) and others (19.8%) including cryptophytes, phytoflagellates, and silicoflagellates. Microphytoplankton was more abundant at eutrophic ST3, with diatoms being dominant taxa while oligotrophic ST2 and ST1 were dominated by nano-scale coccolithophores and dinoflagellates, which is supported by molecular data obtained from the same cruise shown in Durkin et al. (2022). Proportion of taxa read abundances in exported ASVs (Amplicon sequence variants) was distinct between ST2 and ST3, with higher diatom read abundances at ST3 (78%), and lower at ST2 (10%), and small proportion of cryptophyta and haptophyta reads at ST3 (15%), and large contribution at ST2 (49%) (Durkin et al., 2022).

Microphytoplankton

Microphytoplankton abundance increased at eutrophic ST3 (Fig. 2A), where diatoms were dominant (Table 3) and contributed the most to the dissimilarity to other stations (Table 2). Similar assemblage was discussed in a study done by Iriarte & Fryxell (1995) who researched microphytoplankton community structure at equatorial Pacific at 140°W during El Niño 1992 event. Taxa groups that contributed the most to the biomass were diatoms, dinoflagellates and coccolitophores. Dominant species during March to April were Pseudo-nitzschia delicatissima, Thalassionema spp., Thalassiothrix spp., Thalassiosira lineata, and Oxytoxum variabile. In October the same species dominated, with additional two: Calcidiscus leptoporus and Chaetoceros atlanticus. Furthermore, Yamaguchi et al. (2002) analyzed plankton of three regions in western North Pacific: subarctic, subtropical and transitional. Eutrophic subarctic region had the highest phytoplankton biomass, and community dominated with dinoflagellates and diatoms. During North Pacific cruise along 155°W in January 1966, diversity of diatoms was analyzed using light microscopy, and 54 species were identified, of which 37 were frequent (Venrick, 1971). Dominant diatoms were Chaetoceros atlanticus, Ch. peruvianus, Denticula semina e, Nitzschia lineola, Thalassiothrix longissima, and Pleurosigma normanii (Venrick, 1971). Moreover, analyses of time series database including 12-year record of phytoplankton abundances, yielded list of dominant species comparable to ours: N. bicapitata , N. braarudii, N. sicula, Pseudo-nitzschia pseudodelicatissima., Thalassionema sp,.and Oxytoxum cf. variabile (Venrick, 1990). A recent molecular study on diatom assemblages in CCS revealed species with highest relative abundance of reads: Rhizosolenia sp., Actinocyclus sp., Thalassiosira diporocyclus, Asteromphalus sp., and Fragilariopsis doliolus (Abdala et al., 2022).

Eutrophic ST3 had the highest abundances of Pseudo-nitzschia pseudodelicatissima that was absent from ST1 and ST2. Pseudonitzschia taxa, while cosmopolitant (Hasle, 2002), seems to be prevalent in communities along the California coast, where it has been recorded since 1930s (Gran & Thompson, 1930). The appearance of these species coincided with upwelling zones near the coast (Trainer et al., 1998), while others point to areas with increased fertilizer use and agricultural run-off causing eutrophication (Smith et al., 1990). Pseudo-nitzschia species are known to respond to the environmental drivers—both human induced and those specific for ecosystem (Parsons & Dortch, 2002), and increased molecular and taxonomy analyses yielded more knowledge on their ecology, physiology, phylogeny and distribution (Trainer et al., 2012). Species recorded in CCS often causing toxic blooms are P. australis and P.multiseries, while P. pseudodelicatissima was frequent along Washington coast (Trainer et al., 2012). A diatom species Lennoxia faveolata had the second-highest abundance among diatoms at ST3 and was not detected in other stations. (Thomsen et al., 1993) who first described it, found high numbers in samples from Californian waters during winter, but not much more is known about it.

Nanophytoplankton

Horizontal distribution of nano-fraction was even among stations, while cryptophytes abundance increased in eutrophic CCS (Fig. 2B). Molecular analyses by Durkin et al. (2022) confirms this distribution, showing high relative proportion of nano-scale dinoflagellate reads in exported ASVs at both ST2 and ST3 (67% and 78%, repsectively), and high cryptophyta read abundances at ST3 (49%). Furthermore, our results show cryptophytes, coccolithophorids and dinoflagellates dominated entire investigating area (Table 3), and the latter two contributed the most to the dissimilarity of both ST1 and ST2 (Table 2). Similarly, Taylor & Landry (2018) combined epifluorescence microscopy and flow cytometry to assess diversity of North Pacific showing oligotrophic NPSG, and eutrophic CCS are dominated by nanophytoplankton, and micro-scale diatoms, respectively.

Coccolithophores

Coccolithophorid contribution to community composition is significant on all stations, with dominant nano- fraction, especially at greater depths. Micro-scale coccolithophores have a more significant abundance at ST1 and ST2, but they are absent at ST3. Coccolithophorid pigment proxy 19HF has a relatively high ratio on all stations when compared to other pigments. Its presence may point to the higher contribution of pico-scale coccolithophores in bigger depths at ST3. Michaelsarsia adriaticus was a dominant species present only at ST1, whereas dominant Calciosolenia brasiliensis and Ophiaster sp. were specific to ST2. Dominant species observed on both ST1 and ST2 were C. murrayi and Discosphaera tubifera. ST3 was dominated by nano-scale coccolithophorids (ESC <5 µm and 5–10).

Domination of coccolithophores at ST1 and ST2 point to species more adapted to oligotrophic conditions, while indirect observation of 19HF at ST3 implies a shift to the more eutrophic-adapted, smaller coccolithophores species. Li et al. (2013) observed concentrations of 19HF in the Pacific that was low in the upper euphotic zone but increased with depth. Fujiki et al. (2016) also observed low surface 19HF and 19BF concentration (<0.5 mg m−3) that is increasing below 70 m. This would suggest that the coccolithophores are physiologically adapted to low light, nutrient-enriched regions of the water or the 19HF came from other lineages containing the coccolithophorid-indicative marker pigment (Carreto et al., 2001; Landry, 2003).

Okada & Honjo (1973) recorded 90 coccolithophorid species in North and Central Pacific. Based on community structure, they described six zones, and Zones B (Transitional Pacific) and C (Central-North Pacific) match our sampling transect. They observed high abundance of Emiliania huxley (cold variety), Rhabdosphaera clavigera, and Umbellosphaera irregularis. Less abundant species present in this area were Discosphaera tubifera, Syracosphaera spp, R. stylifera, U. tenuis, Umbilicosphaera hulburtiana, U. sibogae, which is similar community structure observed in our study (Table S1), however we observed other dominant species: Calciosolenia brasiliensis, C. murrayi, D. tubifera, Michaelsarsia adriaticus, and Ophiaster sp (Table 3). Hoepffner & Haas (1990) identified nanophytoplankton community of NPSG by using electron microscopy, and observed that Prymnesophyceae contributed the most (55%), with equal abundances of Prymnesiales and Coccosphaerales. Dominant taxa were E.huxley, O. formosu s, R. clavigera, and C. murrayi. Another study in NPSG revealed a total of 53 species, from which most abundant were D. tubifera, U. tenuis, and Heladosphaera cornifera in the upper layer (0–80 m), and Anthosphaera oryza, Florisphaera profunda, Thorosphaera flabellata, and Oolithotus fragilis in the lower layers (140–200 m) (Reid, 1980). Other frequent species observed at the surface were Acanthoica acanthifera, Calyptrosphaera oblonga, U. irregularis, R. stylifera, S. pulchra, and S. pirus (Reid, 1980). Most of listed species we detected as well (Table 3).

Cyanobacteria

In general, Synechococcocus is more ubiquitous (Campbell & Vaulot, 1993; Li, 1995; Blanchot & Martine, 1996; Otero, Álvarez Salgado & Bode, 2020), and often more abundant in colder and nutrient-richer coastal waters (Biller et al., 2014), whereas Prochlorococcus prefers warm oligotrophic waters with temperatures >15 °C (Partensky, Blanchot & Vaulot, 1999), and its abundance drops above 50°N (Partensky, Hess & Vaulot, 1999). Temperature and environmental hydrodynamics may influence variation in the abundances, structure, and distribution of both Prochlorococcus and Synechococcus populations, making them ideal indicator organisms for predicting future changes in the ecosystems caused by the global warming (Babić et al., 2017).

Pigments DVChl a, and zeaxanthin, biomarkers of cyanobacteria Prochlorococcus and Synechococcus, respectively, were recorded in high concentration at oligotrophic NPSG (ST1 and ST2), that falls substantially towards eutrophic CCS (ST3). Zeaxanthin concentrations were less variable throughout investigated transect (Table S4, Fig. 3A). Therefore, our results imply high abundance of Prochlorococcus and Synechococcus at oligotrophic NPSG, with Synechococcous being more adapted to eutrophic ecosystem, which was also confirmed in a study done by Taylor & Landry (2018).

Another research done during Sea to Space Particle Investigation cruise yielded results on the inorganic carbon fixation rates, and nitrate, ammonium, and urea uptake rate at the single cell level in photosynthetic picoeukaryotes (PPE), Prochlorococcus and Synechococcus (Berthelot et al., 2019). The flow cytometry was used, elucidating high abundance of Prochlorococcus at NPSG, and better adaptation of Synechococcus to eutrophic CCS region (Berthelot et al., 2019). Other studies done in oligotrophic regions of the North Pacific also observed dominance of Prochlorococcus, followed by high abundance of Synechococcus (Andersen et al., 1996; Fujiki et al., 2016). These results can be explained with taxa-specific physiological and photosynthetic adaptation to different biogeochemical conditions of the two ecosystems (Partensky, Hess & Vaulot, 1999; Partensky, Blanchot & Vaulot, 1999; Biller et al., 2014).

Synechococcus may also be indirectly observed using the abundance of diatom Leptocylindrus mediterraneus because it has a symbiont colonial protozoan Solenicola setigera Pavillard inside which the Synechococcus may reside (Buck & Bentham, 1998; Gómez, 2007). Leptocylindrus mediterraneus has been detected on both the ST1 and ST2, albeit with low abundance. Nevertheless, the number of cyanobacterial cells should be much higher than the number of symbionts they inhabit. Therefore, although we already detected cyanobacteria by using HPLC pigment analysis, it could be possible to use this indirect three-partner associated symbiosis as a method to record the presence of Synechococcus.

Phytoplankton chemotaxonomy and its relation to microscopy

Chemotaxonomy is a method that allows characterization of the phytoplankton community to coarser taxa than the microscopy can, however, offering insight into the nano- and pico-planktonic composition that is undetectable by classical microscopy methods (Kramer, Siegel & Graff, 2020). Following the decades of research in which pigment composition was related to the microscopy based one, it proved to have biases as concentrations of pigment biomarkers, and their relation to the chlorophyll a are not always the best representative of the targeted taxa (Havskum et al., 2004; Irigoien et al., 2004; Pan et al., 2020). These vary with physiology of the cells, and environmental factors such as irradiance, nutrient availability, day length, temperature, and mixing status (Higgins, Wright & Schluter, 2011).

Regardless of biases, HPLC approach allowed us to track distribution of cyanobacterial taxa indirectly through their pigments proxy, DVChl a, and zeaxanthin, that reached their highest peak at oligotrophic ST1 and ST2. Highest concentrations of fucoxanthin, peridinin, 19HF, alloxanthin, and prasinoaxanthin were observed at ST3 (Table S4), which indicate higher abundances of diatoms, dinoflagellates, coccolitophores, cryptophytes, and prasinophytes respectively. Similarly, chemotaxonomy analyses of subsurface chlorophyll maximum (SCM) at western Baja California also showed dominance of diatoms, prasinophytes, and cryptophytes, in addition to haptophytes, pelagophytes, and picocyanobacteria (Almazán-Becerril, Rivas & García-Mendoza, 2012). Recent HPLC 22-year time series data on CCS phytoplankton community confirms significant contribution of diatoms, followed by dinoflagellates, chlorophytes, prymnesiophytes, and picophytoplankton (Catlett et al., 2021). Furthermore, one interesting trend arose from LINKTREE statistical analysis, showing that pigment alloxanthin determined most the differences in phytoplankton community between the CCS and open ocean stations (Fig. 5). While the imaging-based analysis did point cryptophytes play an important role in distinguishing two communities (Table 2), other taxa, namely coccolithophorids and diatoms also seemed to drive the ecosystem differences.

Claustre (1994) proposed another use of pigments that can determine trophic state of the area by calculating the Fp index (the ratio of pigments highly correlated to changes in Chl a concentration to other pigments that are less variable). Study showed fucoxanthin and peridinin, biomarkers for mostly diatoms and dinoflagellates respectively, had a higher correlation with the change of total Chl a in comparison to other pigments, meaning rise in biomass can be correlated with diatoms and dinoflagellates growth. We used the same approach and calculated higher Fp index at ST3 in comparison to ST1 and ST2, which correlates with the rise of fucoxanthin and peridinin in CCS (Table S4).

Contrasting North Pacific ecosystems

In this paper, the analyzed data showed distinct environments characterized by differences between phytoplankton abundances and concentrations of pigments along a transect that comprises an open ocean and a coast. We recorded lower phytoplankton counts at NPSG which is the largest ecosystem on the planet with reduced intake of nutrients in the euphotic zone, low primary production (PP) and export of carbon to deeper layers (Karl & Church, 2017; Kavanaugh et al., 2018). On the other hand, the phytoplankton counts were higher at the CCS that is more eutrophic ecosystem with seasonal upwellings, higher PP, and frequent blooms (Monterey Bay Aquarium Research Institute, 2022; Closset et al., 2021). Furthermore, the Columbia River influences the CCS with an increased terrigenous contribution raising the trophic state (Hickey & Banas, 2008). Differences between ecosystems can also be detected by observing maximum chlorophyll a concentration, and our results show that it was higher and more variable in CCS, in comparison to NPSG. High chlorophyll a concentrations were already recorded in northern part of CCS (Ware & Thomson, 2005), while studies in other eutrophic ecosystems show similar trend (Zhang, Wang & Yin, 2018; Miranda-Alvarez et al., 2020). Fujiki et al. (2016) observed low Chl a concentration at the ALOHA station (<0.05 mg/m3), whereas at ST1 and ST2 concentration did not exceed 0.4 µg/L.

Conclusions

Our results demonstrate the power of combined techniques, in this case microscopy and pigments, when exploring the ecosystem diversity (Irigoien et al., 2004). We optimized the method for analyzing phytoplankton diversity on a large-scale transect, so with both techniques separately managed to independently differentiating two contrasting ecosystems. Furthermore, each of the techniques, thanks to their strengths and biases, defined different taxonomic drivers. Dominant group revealed by light microscopy were diatoms, with most abundant species being Pseudo-nitzschia pseudodelicatissima. SEM results show cococclithophorids species dominated nanophytoplankton, and their community shifted from large species in NPSG to smaller in CCS. All biomarker signature pigments correlated with taxonomic groups, revealing higher abundance of cyanobacteria in oligotrophic NPSG ecosystem; however, Synechococcus was better adapted to eutrophic CCS in comparison to Prochlorococcus. Cryptophytes were recognized as group with important role in distinguishing between the CCS and NPGS phytoplankton communities, which was confirmed both by pigment and imaging-based analysis. Alloxanthin, zeaxanthin, divinyl chlorophyll b, and lutein were the most important when it comes to distinguishing the community composition across the investigated transect. However, they were not connected to dominating microflora, but to “less charismatic” and elusive microscopy nano- and pico-scale plankton, such as cryptophytes, prasinophytes, and Prochlorococcus. Phytoplankton distinct spatial distribution along the investigated transect indicates variable planktonic dispersal rates and specific adaptations to different trophic ecosystems. Furthermore, observed trends supports findings of other studies indicating a global shift from micro- to nano-scale phytoplankton due to climate-change induced sea surface temperatures rising, causing water layer stratification and oligotrophic environment conditions (Litchman et al., 2007; Finkel et al., 2007; Winder & Sommer, 2012; Yoon & Kim, 2020; Benedetti et al., 2021; Closset et al., 2021). As we are moving towards an era where we will be able to observe global phytoplankton diversity from space (e.g., NASA PACE mission) (Werdell et al., 2019), studies such as these give us an important insight in how different taxonomic approaches (that are used to validate remote sensing algorithms) offer a different views of the changing ocean.

Supplemental Information

Supplemental Information 1 Statistical analysis of phytoplankton abundance data and pigment concentration

(A) SIMPROF global test on Bray–Curtis similarity matrix of phytoplankton abundances. Pi value is 3.685 (vertical dotted line) with significance level of 0.1% (number of permutations is 999. (B) RELATE analysis between phytoplankton abundances and pigment concentrations showing Spearman rank correlation (Rho) of 0.326 (vertical dotted line) with significance level of 0,1% (number of permutations is 999).

Click here for additional data file.

Supplemental Information 2 List of taxa/groups determined by the Utermöhl method and recorded in net samples (20 µm)

Taxa marked with an asterisk were present only in net samples, SEM abbreviation shows species identified with scanning electron microscope.

Click here for additional data file.

Supplemental Information 3 Results of pairwise ANOSIM R test between groups ST1, ST2 and ST3

R statistic shows value between 0 (no difference between ranks) and 1 (difference between ranks). Bold values show groups that are different with statistic significance.

Click here for additional data file.

Supplemental Information 4 BEST Global test. Results show Spearman rank correlation factor for each number of permuted variables

Variable names: alloxanthin (Allo), zeaxanthin (Zea), divinyl chlorophyll b (DvChl b), divinyl chlorophyll a (DVChl a), lutein (Lut), diadino, perdinin (Perid), and diatoxanthin (Diato). Sample statistic (Rho): 0,532. Significance level of sample statistic: 1%. Number of permutations: 99.

Click here for additional data file.

Supplemental Information 5 Pigment concentration (µg/L) averaged by stations

Click here for additional data file.

Supplemental Information 6 Sampling table

Click here for additional data file.

Supplemental Information 7 Species list

Click here for additional data file.

Supplemental Information 8 Phytoplankton counts

Click here for additional data file.

Supplemental Information 9 Chemotaxonomy analyses

Click here for additional data file.

We thank the captain and crew of the R/V Falkor and the “Sea to Space” science party, who made this research possible.

Additional Information and Declarations

Competing Interests

Author Contributions

Data Availability

The authors declare there are no competing interests.

Antonija Matek analyzed the data, prepared figures and/or tables, authored or reviewed drafts of the article, and approved the final draft.

Sunčica Bosak performed the experiments, prepared figures and/or tables, and approved the final draft.

Luka Šupraha performed the experiments, prepared figures and/or tables, and approved the final draft.

Aimee Neeley performed the experiments, analyzed the data, authored or reviewed drafts of the article, and approved the final draft.

Hrvoje Višić analyzed the data, prepared figures and/or tables, and approved the final draft.

Ivona Cetinić conceived and designed the experiments, performed the experiments, authored or reviewed drafts of the article, resources, project administration, funding acquisition, and approved the final draft.

Zrinka Ljubešić conceived and designed the experiments, performed the experiments, authored or reviewed drafts of the article, conceptualization, data curation, resources, supervision, project administration, funding acquisition, and approved the final draft.

The following information was supplied regarding data availability:

The raw data is available in the Supplementary Files.

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
