# Peer review of "Phytoplankton diversity and chemotaxonomy in contrasting North Pacific ecosystems"

_PeerJ, doi:10.7717/peerj.14501_

## Round 0.1 · original submission · Major Revisions

Two reviewers reviewed the manuscript; both agree that the manuscript must be improved in several aspects and suggested that substantial revisions are required.

·

Basic reporting

The manuscript is clear and English language is reasonaly good.

Literature lacked classic and basic backgrounds and references: it was annotated in the general report.

Some figures showed little letter size and makes it difficut to appreaciate. Ra data shares (as supplements).

Some objectives were not fullfilled, as the study of the verticual distribution.

Experimental design

The manuscript has focused obectives and goals, but the sampling design and methods involved were not explained sufficiently.
Rigorous investigation was performed, although results are limited and conclusions are not plausible.
Statistical anaylisis seems OK.

Validity of the findings

Microscopic analysis of phytopankton samples are a remarkable success, as this kind of work is declining. Relevant species found were illustrated by LM and SEM.
The sudy lacks comparisons with previous results back in the 70's and 80's (1900's). Those early papers dealt with microscopic analysis and would yield useful comparisons in the context of climate change.
Impact is limited (as the current ms.) as the conclusions are unimportant.

Additional comments

Evaluation of the ms. titled “Phytoplankton diversity and chemotaxonomy in contrasting
North Pacific ecosystems”, by Matek et al.

This manuscript (ms.) evaluated is based on the analysis of several water samples from only three fixed collecting stations, but a more complicated sampling design. Whereas the results are most interesting and provide important data (most of the phytoplankton data were obtained from microscopic analysis), and the discussion is appropriate to the results obtained, the conclusions are unimportant and very predictable. There is a very complete database and a list of phytoplankton species, and statistical treatments of microscopical and pigments analysis. However, there are some other important points which were not properly covered, such as the lack of details about the vertical distribution (which was stated in the main purposes of the work). The introduction does not include relevant basic references, to understand the evolution on the phytoplankton study of the study area (discussed below). And the methods are not clear regarding samples included in the results provided. No comparison was made between the current results of phytoplankton analyzed by microscopic methods and those obtained in the previous, basic studies (e.g. Venrick and others).
Some questions can be raised, which may improve the potential of the ms.
- Why not using volumes of phytoplankters (by formulas from the literature), for additional comparisons and assessing the importance of each taxonomic group ?

- What samples were used to prepare Tables 2 and 3 ? It is confusing as in Table 1 St1-St3 include a number of samples for every single station (e.g. St1). What samples are considered to be representative ?

- There is no figure regarding the statements mentioned in lines 296-306, about vertical distribution. It would be necessary to see at least one figure on that.

- Comparison of current results with previous observations/results (e.g. Venrick and other, etc.)

There are additional (also important) points such as:

- Some adjectives seem to be redundant and unnecessary (e.g. lines 37, 102)

- There are some typographic errors (e.g. lines 271, 424)

- Line 42- This reference is repeating what was mentioned previously by earlier authors; Otero et al. only reproduce previous information. It is recommended to authors to search for more original references.

- Lines 56-58- Some important references are missing in the backgrounds on the phytoplankton from the north Pacific (e.g. Hayward & McGowan and Venrick, list of some references is given below).

- Line 59- It is not clear if this reference (Almazán-Becerril et al.) is pertinent for the north Pacific.

- Line 70- Reference does not follow citation properly.

- Lines 86, 87- These references are not “recent advances”; they are from 2007 and 2010.

- Lines 96, 97- See comments above (lines 56-58). There are a number of basic and pretty old references on phytoplankton enumerations and microscopic analysis.

- Line 325- There is a more recent reference for Pseudo-nitzschia taxa in the Pacific Ocean (Trainer et al., 2012), and other in the literature.

- Line 338- These implications (important implications) are only based in a few data, but they do not represent temporal variations, and concentrations of all pigments use to vary a lot, even in sort distances. The distributions of picoplanktonic cyanobacteria do also depend on the ecotypes and clades of the two main taxa, Prochlorococcus and Synechococcus. No references added here because there are too many.

- Lines 470-473- Diverse methods for collection and analysis have been using in different regions of the world, including the North Pacific. This conclusion does not look plausible.

- References: Several references missed italics in the scientific names. Lines 573, 646, 659, 664- details of these references are missing.

- Fig. 1. Latitude and longitude of the map LARGER and CLEARER. Fig. 2. Letter size is too small (impossible to see). Fig. 4. Numbers (or letters) to each figure (1, 2, etc., a, b, etc.) (???)

- Table 1, line 4- there is an error.

Unless the authors are able to include or discuss these points in a future revised version, the present ms. is not recommended to be published.


References proposed are:

Hayward TL, McGowan JA. 1985. Spatial patterns of chlorophyll, primary production, macrozooplankton biomass, and physical structure in the Central North Pacific Ocean. J Plankton Res 7: 147-167.
Hayward TL, Venrick EL, McGowan JA. 1983. Environmental heterogeneity and plankton community structure in the central North Pacific. J Mar Res 41: 711-729.

Trainer VL, Bates SS, Lundholm N, Thessen AE, Cochlan WP, Adams NG, Trick C.G. 2012. Pseudo-nitzschia physiological ecology, phylogeny, toxicity, monitoring and impacts on ecosystem health. Harmful Algae 14: 271-300.

Venrick EL. 1971. Recurrent groups of diatom species in the North Pacific. Ecology 52: 614-625.
Venrick EL. 1982. Phytoplankton in an oligotrophic ocean: Observations and questions. Ecol Monogr 52: 129-154.
Venrick EL. 1990a. Mesoscale patterns of chlorophyll a in the Central North Pacific. Deep-Sea Res 37: 1017-1031.
Venrick EL. 1990b. Phytoplankton in an oligotrophic ocean: Species structure and interannual variability. Ecology 71: 1547-1563.
Venrick EL. 1997. Comparison of the phytoplankton species composition and structure in the Climax area (1973-1985) with that of station ALOHA (1994). Limnol Oceanogr 42: 1643-1648.
Venrick EL, McGowan JA, Cayan DR, Hayward TL. 1987. Climate and chlorophyll a: Long-term trends in the central North Pacific Ocean. Science 238: 70-72.

Reviewer 2 ·

Basic reporting

1. Your introduction needs more hydrological details of North Pacific. I suggest that you can move the content of NPSG and CCS at line 107-111 to Introduction.
2. I suggest that you can add latitude and longitude coordinate ruler to Figure 1.
3. At line 134, I suggest you can replace “Four-litre” with “4 L”.

Experimental design

No comment.

Validity of the findings

1. I thank you for providing the raw data, but each CTD cast was deployed at only three depths. It makes the reliability of your data kind of weak. If possible, you can increase the sampling stations and layers to make your conclusion more convincing. Otherwise, the horizontal and vertical distribution of phytoplankton can be influenced by weather, lighting, vertical migration of phytoplankton or other factors. And if you want to discuss ecosystem, you may introduce satellite remote sensing data and ocean current data to support your conclusion of North Pacific ecosystem.
2. I commend the authors for their fieldwork. If there is a weakness, it is in the volume of data. The sampling station was very far away from the other stations, and the CTD deployed depths was limited. The data in this article is enough to clarify the relation between phytoplankton chemotaxonomy and microscopy, but hardly to prove the correlation between phytoplankton community and ecosystem in North Pacific to a certain extent. Under this circumstance, I suggest that you can focus on discussing chemotaxonomy of phytoplankton, but not ecosystem.

---

## Round 0.2 · Minor Revisions

Reviewer #1 indicated the revised version was much improved, but some minor aspects will be addressed before acceptance for publication.

·

Basic reporting

The new version of the manuscript (ms.) evaluated has been considerable improved and I am very satisfied with it. Most or all points raised were dealt with in a proper way. The new version now includes pertinent references for the study area and methods are better explained. Some points are as follows:
There are no more details about the vertical distribution of phytoplankton. It does not matter, but according to the rebuttal letter of the corresponding author, this trait was not dealt with as in another published paper, “very well-mixed layers” were found, which does not necessarily mean that the vertical distribution of phytoplankton should be homogeneous, as shown in Sta1, suppl. Figure 1, where there is a very sharp peak of chlorophyll at about 60 m. depth.
Conclusions: there is a considerable amount of work done for this paper, but the conclusions do not correspond with the main findings; we expect something more than consequences of the methodology employed, for instance the distribution and importance of phytoplankton communities in different environments, or the comparisons with previous papers of the study area (if some important changes could be observed).

And minor points are:
- Lines 280, 283 (PDF doc): Pseudo-nitzschia (complete name of the genus, please)

- Line 347- “inate” this is not well-written and confusing term.

- Lines 429-431: This statement is NOT well-written, but it is additionally very strange.

- Lines 684, 723- why all words are in CAPITALS ?

- Line 753- this reference should be as follows:

Thomsen, H.A., K.R. Buck, D. Marino, D. Sarno, L.E. Hansen, J.B. Østergaard & J. Krupp. 1993. Lennoxia faveolata gen. et sp. nov. (Diatomophyceae) from South America, California, West Greenland and Denmark. Phycologia 32: 278-283.

The work done for this paper is very interesting, it is only matter of adding some important conclusions derived from the results and discussion (authors have all in their hands to do this !) and it will be ready to be published. I hope this evaluation can be useful.

Experimental design

The new version now includes pertinent references for the study area and methods are better explained.

Validity of the findings

There are no more details about the vertical distribution of phytoplankton. It does not matter, but according to the rebuttal letter of the corresponding author, this trait was not dealt with as in another published paper, “very well-mixed layers” were found, which does not necessarily mean that the vertical distribution of phytoplankton should be homogeneous, as shown in Sta1, suppl. Figure 1, where there is a very sharp peak of chlorophyll at about 60 m. depth.

Additional comments

Evaluation of the ms. titled “Phytoplankton diversity and chemotaxonomy in contrasting
North Pacific ecosystems”, by Matek et al.

The new version of the manuscript (ms.) evaluated has been considerable improved and I am very satisfied with it. Most or all points raised were dealt with in a proper way. The new version now includes pertinent references for the study area and methods are better explained. Some points are as follows:
There are no more details about the vertical distribution of phytoplankton. It does not matter, but according to the rebuttal letter of the corresponding author, this trait was not dealt with as in another published paper, “very well-mixed layers” were found, which does not necessarily mean that the vertical distribution of phytoplankton should be homogeneous, as shown in Sta1, suppl. Figure 1, where there is a very sharp peak of chlorophyll at about 60 m. depth.
Conclusions: there is a considerable amount of work done for this paper, but the conclusions do not correspond with the main findings; we expect something more than consequences of the methodology employed, for instance the distribution and importance of phytoplankton communities in different environments, or the comparisons with previous papers of the study area (if some important changes could be observed).

And minor points are:
- Lines 280, 283 (PDF doc): Pseudo-nitzschia (complete name of the genus, please)

- Line 347- “inate” this is not well-written and confusing term.

- Lines 429-431: This statement is NOT well-written, but it is additionally very strange.

- Lines 684, 723- why all words are in CAPITALS ?

- Line 753- this reference should be as follows:

Thomsen, H.A., K.R. Buck, D. Marino, D. Sarno, L.E. Hansen, J.B. Østergaard & J. Krupp. 1993. Lennoxia faveolata gen. et sp. nov. (Diatomophyceae) from South America, California, West Greenland and Denmark. Phycologia 32: 278-283.

The work done for this paper is very interesting, it is only matter of adding some important conclusions derived from the results and discussion (authors have all in their hands to do this !) and it will be ready to be published. I hope this evaluation can be useful.

Reviewer 2 ·

Basic reporting

This is a review report for the manuscript, entitled “Phytoplankton diversity and chemotaxonomy in contrasting North Pacific ecosystems” (Manuscript number: #74249). In this study, phytoplankton diversity and chemotaxonomy in the surveyed sea area were analyzed. Overall, I think the workload of this study is insufficient and the quality of manuscript writing needs to be improved. The vertical distribution mentioned by the reviewer before, I also think it is very important. The author explained that "the water layer is well mixed", so the vertical distribution is not required. I don't think it can convince me.

Below are some specific comments that I hope will be helpful to the manuscript.

1. Title:There are only 3 survey stations in this study, all of which are located in the nearshore area, and the research results show that many nearshore species are dominant species, which does not match the previous large-scale sampling results of the Pacific open stations. I think it cannot represent the North Pacific. Too broad description of the survey area will mislead readers. The author should describe the survey area more accurately.

2. Abstract: Re-write this part.
Background: This study did not involve phytoplankton and changes, anthropogenic pressure and their impact on the oceans. The background is only to express the importance of phytoplankton, but does not reflect the significance of this study under global change.
Methods: Only the sampling area is written, and there is no simple description of what method was used.
Results: It is only a brief comparison of the two ecosystems (NPSG CCS), without highlighting the main research results of this study, and has no reference significance.
Conclusion: All are known conclusions and do not reflect the innovation of this study.

3. Line 55: “baseline” This word is inaccurate, and it is recommended to replace it.

4. Line 62-69: The references are old and the phytoplankton community structure may have changed during this period. It is recommended to keep the existing ones and add citations to newer references.

5. Line 81-83: “eutrophic and nutrient supply decreased for 50 %?” Does the author want to express more or less nutrients?

6. Line 93-94: This is true, but this study is not a long-term study, such a statement would like to explain the inadequacy of this study?

7. Line 95-99: Molecules, imaging techniques, and chemotaxonomy, providing perspectives and insights. Such a statement is too simplistic and does not reflect the meaning of the author's expression of such a point of view.

8. Line 104: I don't think the classic method disciplines are in decline.

9. Line 108: This study did not address the effects of warming on phytoplankton.

10. Line 111: I don't think this study is on a large spatial scale.

11. Line 113: Just write what this research covers.

12. Line 123: “CTD” When it first appears, write the full name, then abbreviated.

13. Line 152-155: That is, only the phylum was identified, and the species level was not identified. Although the taxonomy is complex, studies identifying it down to the species level are common.

Experimental design

no comment

Validity of the findings

no comment

Additional comments

no comment

---

## Round 0.3 · accepted · Accept

The authors have completed the minor changes.